# Performance of Fluorescence and Diffuse Reflectance Hyperspectral Imaging for Characterization of Lutefisk: A Traditional Norwegian Fish Dish

**DOI:** 10.3390/molecules25051191

**Published:** 2020-03-06

**Authors:** Abdo Hassoun, Karsten Heia, Stein-Kato Lindberg, Heidi Nilsen

**Affiliations:** Nofima, Norwegian Institute of Food Fisheries and Aquaculture Research, Muninbakken 9-13, 9291 Tromsø, Norway; karsten.heia@nofima.no (K.H.); stein-kato.lindberg@nofima.no (S.-K.L.); heidi.nilsen@nofima.no (H.N.)

**Keywords:** fish, fluorescence, diffuse reflectance, lutefisk, online measurements, spectroscopy

## Abstract

Lutefisk is a traditional Norwegian fish dish made from dried fish, such as cod or other whitefish. In Norway and other Nordic countries, lutefisk is considered among the most popular dishes served during Christmas or other festive occasions. However, to date, little attention has been paid to this product, and available research on the quality, processing, and chemistry of lutefisk is still limited. The quality of this very delicate product, with a high pH value, depends on many factors, such as the initial quality of raw materials (stockfish), the quantity of lye used during the preparation process of lutefisk, and time during soaking in the lye and water, among others, making it challenging to both optimize processing and monitor the quality of lutefisk. In this study, four commercially available lutefisk brands (labelled as A, B, C, and D) were characterized using two online spectroscopic techniques, namely fluorescence and diffuse reflectance hyperspectral imaging, implemented on conveyor belts to mimic industrial applications. The samples were also analyzed by the use of an offline laboratory instrument based on visible/near infrared diffuse reflectance spectroscopy. Three traditional measurements, including texture, water content, and pH, were also conducted on the same samples. Supervised classification PLS-DA models were built with each dataset and relationships between the spectroscopic measurements and the traditional data were investigated using canonical correlations. The spectroscopic methods, especially fluorescence spectroscopy, demonstrated high performance for the discrimination between samples of the different brands, with high correlations between the spectral and traditional measurements. Although more validations of the results of this study are still required, these preliminary findings suggest that the destructive, laborious, and time-consuming traditional techniques can be replaced by rapid and nondestructive online measurements based on hyperspectral imaging used in fluorescence or diffuse reflectance mode.

## 1. Introduction

Lutefisk is a traditional fish dish made from air-dried cod, or other whitefish, and marinated in lye and water for several days before it is cooked. Norway and some other Nordic countries have a long history with lutefisk and this dish is one of the traditional meals served during Christmas. Lutefisk is also popular among the descendants of Scandinavian immigrants in the United States. 

Nordic cuisines are surprisingly diverse, with a variety of fish dishes prepared in such a way that complex processes of biochemical transformations and microbial changes take place [1,2,3]. Lutefisk is one of these dishes whose preparation method is particularly special and uncommon. The name “lutefisk” literally means “lye fish”, referring to the main particularity and the most distinguished stage of preparing lutefisk, which is the marinating in lye. In more details, the seasonal winter and spring cod (locally called *skrei*) caught in Northern Norway is hanged on drying racks for several months to dry in the wind and sun to produce so-called stockfish. For production of lutefisk, the stockfish is soaked in cold water for about a week, which allows the fish to be rehydrated. Then it is immersed in a lye solution (caustic soda) for two to four days, reaching a pH of 11–12. The last step is to soak the fish again in water in order to remove the lye and make the fish edible. Due to the long period of soaking in water and lye, the fish gains a soft, sometimes jelly-like consistency.

Although lutefisk was mentioned for the first time in the literature by the Nordic priest and ethnographer Olaus Magnus in the 16th century [4], very little scientific work has been carried out on this fish dish. In a recent study, attention has been given to the role of microbiota in lutefisk [5]. However, studies investigating relationships between the quality of stockfish and the final product, different preparation methods and impact of length of soaking time and quantity of lye on the sensory quality, chemical composition, and nutritional quality of lutefisk, remain to be conducted. 

Quality of fish can be analyzed by a wide range of traditional methods, including sensory, microbiological, and physicochemical analysis. Although considered as reference methods, most of these techniques are costly, laborious, time-consuming, and destructive, and therefore, cannot be applied online during production. On the other hand, the rapidity and non-destructive nature of measurements obtained by spectroscopic techniques, such as fluorescence spectroscopy and visible/near infrared spectroscopy, have become a driving force to encourage the food industry to move away from offline/at-line quality assessments to online/inline processes [6,7]. Fluorescence spectroscopy has shown its potential in many applications, such as monitoring quality changes in frozen horse mackerel (*Trachurus japonicus*) fillets [8] and spotted mackerel (*Scomber australasicus*) fillets [9], investigating the impact of different cooking methods of hairtail (*Thichiurus lepturus*) fillets on the formation of Maillard reaction products [10], and demonstrating the effect of different storage conditions on the freshness of whiting (*Merlangius merlangus*) fillets [11]. Visible/near infrared spectroscopy has been widely applied for a wide range of agricultural products due to many desirable features, such as rapid preparation and analysis of samples, low cost, and non-destructive measurements [7]. Performing the measurements in the diffuse reflectance mode provides many advantages, such as homogenous illumination of the sample without shadow on the surface or the background material, thus enabling robust and reliable spectroscopic results to be obtained [6]. Recent applications of diffuse reflectance spectroscopy included the estimation of the amount and distribution of blood in whitefish fillets [12] and quantification of met-myoglobin proportion and meat oxygenation in pork and beef [13]. Hyperspectral imaging has emerged in recent years as a combination of two techniques, namely imaging and spectroscopy in one measurement, providing both spectral and spatial information to be obtained at the same time. Although the use of hyperspectral imaging in the visible/near infrared has been widely investigated [12,14], little research has been found in the literature examining the effectiveness of hyperspectral imaging in the fluorescence mode [15]. The objective of this study was to investigate the feasibility of two online spectroscopic methods, namely fluorescence and diffuse reflectance, coupled with hyperspectral imaging, and one offline spectroscopic technique, based on the visible/near infrared spectral region, for the characterization of four commercially available brands of lutefisk. Traditional analyses of water content, pH, and texture were also conducted on the same samples in order to characterize the products by traditional methods. 

## 2. Results and Discussion

### 2.1. Traditional Measurements

#### 2.1.1. Water Content and pH

Water content (WC) is one of the main quality parameters of fish, which affects the growth of microorganisms, thus influencing the quality, safety, and shelf life of fish [16]. WC in lutefisk is even a more important factor in the case of lutefisk due to the very high contents of water in this product. Higher WC means more bacterial growth, which, in turn, results in a shorter shelf life as the WC is a critical factor for microbial development. 

Our results show that the WC in samples of the examined lutefisk brands varied from 86.9 to 96.8%. The ANOVA followed by Tukey test show significant differences between the brands A and D and between brands C and D, with the WC values being significantly higher in brands A and C (Figure 1). A significantly lower WC value was also observed in brand B compared to brand C. Differences in WC can be due to variations between the different brands regarding fish to water ratio and the length of time during which the fish is soaked in the water during preparation of lutefisk. Moreover, stockfish used to prepare lutefisk can contain different amounts of water as a function of the size of fish as well as drying parameters applied during the drying process [17], which, in turn, can be reflected in various levels of WC in lutefisk final products.

Changes in pH values are commonly used as a complementary tool in evaluating the quality of fish and other seafood. However, this parameter is of great importance for lutefisk as it is related to the quantity of lye employed during the preparation of lutefisk and the duration of the marinating process. The values of pH of the four brands of lutefisk were different and varied from 9.83 to 10.87. The average of pH values of the four brands showed that brands B and D had the lowest and the highest values, respectively (Figure 1). The ANOVA test displayed that brand B had a significantly lower pH value as compared to the other lutefisk brands, despite the relatively high variability within this brand with a large standard deviation value. No significant difference was observed between the three other lutefisk brands. 

#### 2.1.2. Texture Parameters

Soaking fish in water and lye gives lutefisk a soft and somewhat jelly-like texture. Thus, controlling the texture and consistency of lutefisk is of the utmost importance in order to optimize quality and acceptability of this product. Six texture parameters were obtained from a Texture Profile Analysis (TPA).

The hardness refers to the maximum force during the first compression cycle. Brand A presented significantly higher hardness values compared to brands B and C, with these latter being not significantly different from each other. A higher but not significant difference was observed for the hardness of brand A as compared to brand D. The adhesiveness refers to the negative force area under the baseline between the compression cycles, representing the work necessary to overcome the force of attraction between the sample and the probe surface. Significantly higher adhesiveness values were observed for brands A and C compared to the two other brands, probably due to their higher WC. The springiness describes the ability of fish muscle to regain shape when the deforming stress is removed, while the cohesiveness represents the resistance of the muscle during the deformation. These two texture parameters were significantly higher in brand C compared to the other brands. A similar trend was also observed for the resistance values, as brand C had significantly higher values compared to the other brands. As the chewiness value is calculated from the hardness, this parameter had the same trend as the hardness, with brand A having significantly higher chewiness values compared to the other brands. The results of the most texture parameters are not in agreement with the other traditional measurements, suggesting that other parameters than the WC and pH would probably affect the texture of lutefisk. The texture of the final product of lutefisk could be affected by many other processing parameters, such as lye concentration, soaking duration, etc.

### 2.2. Spectral Features

#### 2.2.1. Online Fluorescence Hyperspectral Imaging Spectra

Fish can be considered as a multifluorophoric matrix due to its content of several fluorescent compounds (called fluorophores), such as nicotinamide adenine dinucleotide (NADH), amino acids, riboflavin, oxidation products, and collagen. Fluorescence spectroscopy is very sensitive to changes in the local molecular environment of fluorophores, and several factors (pH, temperature, polarity, color, etc.) in the food matrix highly influence fluorescence signals [18,19,20,21]. 

Raw spectra of fluorescence emission, obtained from all samples of the four brands, are shown in Figure 2a. The spectra of the four brands showed a similar pattern with a broad peak around 450 nm, whereas differences in the fluorescence intensity were observed, probably due to the various levels of WC or pH values of the four brands, as shown previously. Differences in color between the four lutefisk brands can also result in different fluorescence properties. Little work has been reported in the literature on the impact of these parameters on fluorescence spectra. For example, a relationship was noticed between fluorescence intensity of red sea bream (*Pagrus major*) and water holding capacity and water loss during storage [22], while a connection between a shift in the fluorescence emission spectra and changes in pH in a myofibrillar model system was suggested [23].

A decrease in the fluorescence intensity of these four brands was observed compared to fluorescence intensities of raw samples (without cooking) of the same brands (data not shown). This decrease is probably due to a decrease in collagen content and degradation of connective tissue induced during cooking. Other data in the literature demonstrated significant changes in fluorescence intensity in cooked fish as compared to raw samples [10]. A trend of higher fluorescence intensities can be seen for most of the samples of the brands packed under vacuum (B and D) compared to those packed without vacuum, probably due to reduced quenching effects in the absence of oxygen [24].

However, it should be stressed that the fluorescence experiment in this study was conducted based on a single excitation wavelength, which might limit the possibility of detecting fluorescence arising from several fluorophores in lutefisk at the same time. Therefore, future work should evaluate lutefisk over a whole range of emission spectra at different excitation wavelengths by using fluorescence excitation–emission matrices, called fluorescence landscapes, thus creating three-dimensional maps for excitation, emission, and fluorescence intensity. 

#### 2.2.2. Online Diffuse Reflectance Hyperspectral Imaging Spectra

The reflection intensity of the online hyperspectral imaging diffuse reflectance spectra (400–1000 nm) obtained on the four brands of lutefisk is shown in Figure 2b. These spectra are characterized by high reflectance values, indicating a high amount of water in the measured samples [25]. This agrees with the measurements of WC, which displayed values higher than 95% in certain brands. A weak absorbance can be seen around 675 nm, while the absorbance peak around 980 nm is due to the presence of O–H stretching second overtone of water [14]. Due to their higher content of water, most samples of the brand C had higher absorbance values (lower reflectance) compared to the other groups. The differences between spectral reflectance values can be related mainly to the water content. In addition, differences in color of the different lutefisk brands, likely due to variations in raw materials (stockfish) used for preparing lutefisk, could cause variations in spectral reflectance values. Our results also indicate that cooked lutefisk samples had lower reflectance values than those of raw samples. The spectral changes induced by thermal treatments could be due to denaturation of proteins and modifications in water states in fish, resulting in changes in light absorbance and scattering intensities [26,27].

#### 2.2.3. Offline Visible/Near Infrared Diffuse Reflectance Spectra

Raw spectra of absorbance in the visible/near infrared region (400–2500 nm), obtained from all samples of the four brands, are shown in Figure 2c. Characteristic bands of samples with higher contents of water can be observed at around 980, 1200, and 1450 nm due to the O-H stretching second overtone, C-H stretching second overtone, and the O-H stretching first overtone, respectively, while the region between 1800 and 2200 nm is characterized by combinations of O-H and N-H stretching [28,29,30]. Again, most of the samples of brands C and A showed higher absorbance values, probably due to their higher WC compared to the two other brands. Other bands related to proteins and fats cannot be seen from these spectra because of the very low content of these components in lutefisk, in addition to the broad bands of water that could overlap with weaker peaks.

### 2.3. Multivariate Analysis 

The spectra obtained from the different brands have a similar pattern and cannot be easily distinguished (Figure 2). Hence, the application of a multivariate analysis including unsupervised (e.g., PCA) and supervised (e.g., PLS-DA) techniques makes it possible to extract useful information from such complex spectra. 

#### 2.3.1. Preliminary PCA on the Spectral Data

In a first step, a PCA was applied to each dataset to display inherent systematic patterns of variation between the brands, and score plots resulting from the PCA applied to the fluorescence and diffuse reflectance spectra are shown in Figure 3. The first two PCs of the PCA applied to the fluorescence dataset showed a clear separation between samples of the four lutefisk brands (Figure 3a). According to the first PC, most of the samples of brands A and C had positive score values, while samples of the brands B and D were located close to each other on the negative side of the first PC. These results can be explained by the low content of water in brands B and D compared to brands A and C, or by the type of packaging, as the former brands were packed under vacuum, whereas the latter were packed without vacuum. Other studies in the literature reported changes in fluorescence spectra as a function of packaging type, i.e., the presence/absence of oxygen [19,24]. The PCA loadings for PC1 (results not shown) demonstrate that the main wavelengths responsible for the distinction of the different brands of lutefisk were located around 450–470 nm. According to the literature, several fluorophores could be responsible for the fluorescence around these bands, even though the NADH often has been reported to be the main fluorescence substance.

Similar results were obtained with the PCA applied to the online diffuse reflectance dataset (Figure 3b). Here, the first and the second PCs explained a large amount of variance (96.77%). Again, the brands of high content of water (i.e., A and C) were separated from each other and from lutefisk brands of low water contents, whereas some overlapping was observed for the brands B and D. Less grouping of samples of the four brands was observed based on the PCA applied to the offline visible/near infrared diffuse reflectance spectra (Figure 3c). This result can be attributed to the high content of water in lutefisk, as water is known to have a significant impact on the NIR infrared spectra. Indeed, water absorbs strongly in the NIR region, and thus, has broad absorption bands that can mask weaker absorption features. For this reason, some authors used preprocessing methods to increase resolution of the spectra and reduce scattering [26], whereas others avoided completely the NIR region in their modelling [27,31]. In this study, the use of preprocessing techniques did not significantly improve the results. However, the application of PCA to the spectral range between 400 and 1000 nm gave better results, with a similar performance as the online diffuse reflectance spectra. The score plot of the PCA defined by the first two PCs revealed that the four groups of lutefisk could be separated quite well from each other (Figure 3d). According to the first PC, most samples belonging to the brands A and C had positive coordinates, while samples of the brands B and D had mostly negative coordinates. 

#### 2.3.2. PLS-DA for Discrimination and Classification Purposes

In a second step, the PLS-DA model was applied to the spectroscopic measurements in order to investigate the discriminant ability of each spectral data set. 

The results of PLS-DA applied to the fluorescence data are shown in Figure 4. Cross-validated predicted classes (brands) and some statistics referring to the performance of the model are presented in Figure 4a. It can be noticed that for the four lutefisk brands, only one color referring to one class (brand) can be seen above the red dotted line, on the contrary to most samples of the other brands, indicating a good performance of the model. Indeed, a perfect separation can be noticed for classes A and C, whereas only one sample from class C and D were misclassified and attributed to class B, and four samples were misclassified in the case of class D. In addition, optimal sensitivity values of 1.00 for brands A, B, and C, and a value of 0.94 for brand D were observed, while the specificity values were 1.00, 0.96, 1.00, and 0.94 for brands A, B, C, and D, respectively. Error rates, calculated for each sample and then averaged over all samples, were of 0.014, 0.001, 0.042 and 0.028 for the brands A, B, C, and D, respectively. 

Figure 4b corresponds to ROC curves (on the left-hand side) made by plotting the sensitivity versus the false positive values (1−specificity). For a stable and robust PLS-DA model, the ROC curve hits the upper left corner, which indicates no false positives and no false negatives, giving a high area under the ROC curve value (AUC). For brands A and C, the red points on the ROC curves are situated exactly in the upper left corner, giving a perfect AUC values, whereas these points are very close to the upper left corner and gave AUC close to one in the case of B and D brands. 

ROC plots on the right-hand side of the Figure 4b show similar information in a different format; the plots represent the choice of the decision threshold to obtain optimal combination of sensitivity and specificity values. The point at which sensitivity and specificity meet is the balance between false positives and false negatives, and the higher up the *y*-axis this point is, the better the model fits the data. In the case of A and C brands, the blue line (specificity values) goes to the top until the values of one and does not cross the red line (sensitivity values), providing strong evidence that this model is robust and promising, while close to perfect results can be noticed for the B and D brands. Moreover, low misclassification errors of 1%, 0%, 4%, and 2% were obtained for brands A, B, C, and D, respectively, with an overall correct classification rate of 91.43%.

Similar or slightly lower performances were observed for the PLS-DA model applied to the online hyperspectral imaging diffuse reflectance data (Figure 5). Only a few samples were misclassified, with high values of sensitivity and specificity being observed for the four lutefisk brands (Figure 5a). Indeed, the sensitivity values ranged between 0.94 for brand C and D and 1.00 for brand A and B, while specificity values of 0.96 for brands A and B, and 0.98 for brands C and D were obtained. Classification errors of 0.01, 0.01, 0.00 and 0.07 were obtained for the brands A, B, C and D, respectively. Additionally, high AUC values (equal or close to 1) were obtained, and the plots of sensitivity versus specificity computed across a range of threshold values showed high performances (Figure 5a). Moreover, an overall correct classification rate of 90% and misclassification errors ranging from 0 to 7% were achieved.

The PLS-DA model applied to the whole spectral range (400–2500 nm) of the offline visible/near infrared diffuse reflectance data displayed lower rates of correct classification, lower values of sensitivity and specificity, and higher misclassification errors. Better results were achieved by applying the PLS-DA model to the 400–1000 nm spectral range. In this case, sensitivity and specificity values were 0.88, 0.94, 0.94, 0.89, and 0.81, 0.92, 0.94, 0.88 for brands A, B, C and D, respectively. AUC values of 0.95, 0.98, 0.97 and 0.94 were obtained for brands A, B, C and D, respectively.

Misclassification errors were 7%, 9%, 13%, and 17% for brands A, B, C, and D, respectively, with overall correct classification rate of 64%. 

The overall results of the discriminant models show that fluorescence spectroscopy outperforms the other spectroscopic techniques used in this study, probably due to its high sensitivity and specificity compared to other spectroscopic techniques [19,21]. 

#### 2.3.3. Relationships between Spectroscopic and Traditional Data

In order to assess the capability of spectroscopic techniques to act as monitoring tool, it is important to consider correlations between spectral data and traditional measurements. Thus, in a third step, CCA was applied to investigate correlations between two sets of variables; one set comprised one spectral dataset (online fluorescence hyperspectral imaging, online diffuse reflectance hyperspectral imaging, or offline visible/near infrared diffuse reflectance data) and the other one comprised traditional data (one vector of all traditional measurements). 

First, CCA was applied to the fluorescence emission spectra (410–570 nm) as the X-variables set and the traditional data (i.e., WC, pH, and six texture parameters) as the Y-variables set. The results demonstrate that these two datasets were highly correlated. Small Wilks’ lambda and *p* (<0.001) values were obtained on first two pairs of canonical variables. The canonical correlation coefficients of the first two pairs of the canonical variables were 0.997 and 0.994 (Figure 6a). This result illustrates that the traditional data can be derived from the fluorescence measurements [32,33]. Canonical loadings were used as a criterion for interpreting the relative importance of each of the initial variables (results not shown). The results pointed out that, from the fluorescence data (X-variables set), the wavelengths around 460 and 510 nm contributed the most in acquiring the canonical variables, while texture parameters, including springiness, adhesiveness, hardness, and chewiness were the most important variables from the traditional data (Y-variables set). 

Similar results were obtained from the CCA applied to examine the correlation between the online diffuse reflectance hyperspectral imaging data and the traditional measurements. The first two canonical variables calculated from the diffuse reflectance data and the traditional measurements gave high squared canonical correlation coefficients equal to 0.994 and 0.986 (Figure 6b). Again, the result indicates the high efficiency of the diffuse reflectance data for predicting these traditional measurements. The canonical loadings showed that from the diffuse reflectance data, the visible range (especially the wavelengths between the 500 and 600 nm) were the most important in acquiring the canonical variables, while from the traditional data, the cohesiveness parameter contributed the most in the canonical variables. 

As the number of the variables should be small relative to the number of samples, in the case of the offline visible/near infrared diffuse reflectance data, the first ten PCs of the PCA were used as the X-variables set, while as described previously, the Y-variables set is the traditional data. The results show squared canonical correlation coefficients equal to 0.854 and 0.746 for the first two pairs of canonical variables (results not shown). The canonical loadings showed that texture parameters, especially the cohesiveness, contributed the most in the canonical variables, in agreement with the previous results obtained from the online measurements. 

Springiness, cohesiveness and adhesiveness are important texture parameters for lutefisk. This was confirmed in this study by their high contributions to the model. Thus, it can be concluded that the same information obtained from tedious, laborious, and offline measurements can be acquired rapidly from a non-destructive online analysis. 

## 3. Materials and Methods

### 3.1. Preparation of Lutefisk Samples and Cooking Procedure 

Four brands of frozen lutefisk, labelled as A, B, C and D were purchased from a local market in Tromsø (Norway). Lutefisk of the four brands was made from cod, and two of these brands (B and D) were packed under vacuum, whereas the two others (A and C) were packed in normal atmosphere. The fish was thawed overnight at 4 °C and then cut in small pieces and cooked in a traditional oven at 200 °C for 40 min. This cooking method was employed as a representative of conventional, domestic, and most commonly used cooking methods for lutefisk. After cooking, samples were kept in a cold room (humidity 80–90%, temperature 4 ± 2 °C) for one hour, then cut and placed in small glass dishes for measurements. Eighteen (18) samples from each brand were cooked and analyzed, while two samples were analyzed without cooking (raw), meaning that a total of 20 samples from each brand were analyzed. The spectroscopic measurements were performed first, starting with the online fluorescence spectroscopy followed by the online diffuse reflectance analysis and the offline visible/near infrared diffuse reflectance measurements. Then, the measurements of the texture, pH, and water content were performed on the same samples. 

### 3.2. Traditional Measurements

Water content (in %) was determined as the difference in weight before and after drying samples (4.17 ± 2.19 g) in an oven at 105 °C during 24 h. The pH value was measured directly on the sample (Weilheim, WTW, pH330i, Germany) or in liquid (Mettler Tole-do, Schwerzenbach, Switzerland) according to the manual of the producers. The mean value of the two measurements was then calculated. It is believed that this combination of measurements on intact muscles and ground samples could work best for the pH measurements. Texture profile analysis (TPA) was performed using the TA.HDplus texture analyzer (Micro Stable System) at room temperature. The test conditions were the following: two consecutive cycles of 50% compression using a flat-ended cylindrical probe (TA-40A), pre-test speed: 5 mm/s, test speed: 1 mm/s, post-test speed: 10 mm/s, target mode: strain, trigger force: 5 g. The following TPA parameters were obtained from force–time curves: hardness, adhesiveness, cohesiveness, springiness, chewiness, and resistance.

### 3.3. Spectroscopic Measurements

#### 3.3.1. Online Fluorescence Hyperspectral Imaging 

Fluorescence measurements were carried out using a pushbroom hyperspectral camera (Norsk Elektro Optikk, model VNIR-1024) operating in the spectral range from 400–1000 nm with a spatial resolution of 0.28 mm across-track and 0.48 mm along track. A focused LED UV line light was used for the excitation with a centre wavelength of 365 nm (Metaphase UL-LL409-UV365-24) and the emission was recorded at wavelengths higher than 400 nm. The camera was fitted with a lens focused at 1000 mm, mounted 1020 mm above the conveyor belt carrying the lutefisk samples to be imaged. Three-dimensional hyperspectral image cubes, built line by line, were obtained by moving the conveyer belt at a speed of 1 cm/s.

#### 3.3.2. Online Diffuse Reflectance Hyperspectral Imaging 

Diffuse reflectance measurements were performed using a VNIR-640 imaging camera (Norsk Elektro Optikk, Skedsmokorset, Norway) with a spectral range of 430–1000 nm. The camera was mounted 1020 mm above a conveyor belt carrying the lutefisk samples and moving at a speed of 40 cm/s. The samples were illuminated with a custom-made light source, which consists of 14 halogen bulbs (50 W) mounted inside a box consisting of 10 mm thick high-density polytetrafluoroethylene (PTFE, also known as Teflon) plates. The measuring setup is thoroughly described elsewhere [12]. 

#### 3.3.3. Offline Visible/Near Infrared Diffuse Reflectance Measurements

Visible/near infrared diffuse reflectance spectra were recorded at 0.5 nm intervals using an XDS Rapid Content Analyzer (FOSS NIR Systems, Inc., Laurel, MD, USA). The samples were placed in a 35 mm quartz cuvette and scanned at room temperature in the wavelength range of 400–2500 nm. For each sample, an average of 32 scans with a field of view diameter of 17.25 mm was recorded. The spectra were collected using Vision software packages (NIR-Systems, Silver Spring, MD, USA) and stored in optical density units log (1/R), where R represents the fraction of incident light that is reflected.

### 3.4. Chemometric Analysis 

Extraction and all processing of hyperspectral data were performed in IDL 8.6 (Exelis Visual Information Solutions, Bracknell, United Kingdom). Regions of interest were manually selected from the center of the image of each sample and the average spectral data were generated from the images. Analysis of variance (ANOVA) and Tukey’s Honestly Significant Difference (HSD) post-hoc test performed in the R software, were used to test differences between sample means, which were considered significant at *p* < 0.05. Principal Component Analysis (PCA) was applied to each spectral dataset in order to detect possible patterns in the data. Then, the discriminant ability of each spectral dataset was determined by applying Partial Least Square Discriminant Analysis (PLS-DA) models. The PLS-DA is one of the most powerful and commonly used discriminant classification models, especially for spectroscopic data [29]. Moreover, this classification method is fast compared to other classifications techniques (e.g., support vector machine). PLS-DA predicts membership of an individual to a qualitative group preliminary defined. Strict class prediction rule based on assignment of a sample to a class if it has probability above threshold (>50%) for that class only was used. PLS-DA models were optimized using less than 20 latent variables, cross-validated with 10-fold venetian blinds including seven samples per split, and the data were standardized using the autoscaling method (centralization and normalization). Several criteria could be used to evaluate the performance of PLS-DA models [28,34]: *Sensitivity* refers to the proportion of positive cases that are correctly identified. *Specificity* provides the proportion of negatives cases that are classified correctly. *Misclassification error* represents the proportion of samples which are incorrectly classified. *Correct classification rate* is the number of correctly classified samples divided by the total number of samples. *Receiver Operator Curve (ROC)* is another valuable tool that can be used to evaluate the performance of the PLS-DA model, displaying the relationship between the sensitivity and the specificity. The ability of different PLS-DA models is compared using the *Area Under the Curve (AUC)*, which should be one or close to one for robust models. The analyses were performed by using PLS-Toolbox v.8.5 (Eigenvector Research) for MATLAB R2018a. 

Finally, Canonical Correlation Analysis (CCA) was used to describe the correlations between two sets of variables X and Y obtained on the same samples, meaning that the number of observations (rows) must be the same, but can have different numbers of variables (columns). This method generates a new set of variables, called canonical variables, so that the highest correlation can be achieved between a linear combination of the variables in the first dataset and a linear combination of the variables in the second dataset [32,33]. The CCA was carried out using MATLAB R2018a.

## 4. Conclusions

This study was conducted to investigate the potential of spectroscopic methods to characterize lutefisk rapidly and in a nondestructive manner as opposed to traditional measurements. As most spectroscopic studies have been conducted with offline measurements using laboratory spectroscopic instruments, this investigation was intended to illustrate whether fluorescence and diffuse reflectance spectroscopic techniques can be applied on moving samples on conveyor belts in order to mimic industrial environments, with a detector being placed above the conveyor belt without any physical contact between the scanned samples and the measuring setup. 

The best classification model was obtained with the online fluorescence spectroscopy followed by the online diffuse reflectance spectroscopy, while a less performance was observed with offline visible/near infrared diffuse reflectance spectroscopy. Canonical correlation analysis applied to the traditional and the spectral data showed high performance on fluorescence data. Therefore, it can be concluded that fluorescence hyperspectral imaging technique has a great potential to become a useful quality control method for rapid online analysis of lutefisk. However, the speed of the conveyer belt should be increased to meet the industrial production requirements. Moreover, much effort should be put into determining the most effective excitation wavelengths, thus recognizing the relevant fluorophores in lutefisk, by using fluorescence landscapes instead of excitation at a single wavelength. As our primary focus on this paper was put on the feasibility of the methodology developed for lutefisk, more validation of these results should be carried out in more extensive studies. 

To the best of our knowledge, this is the first study on spectroscopic characterization of lutefisk. Our next work will focus on the use of spectroscopic methods for prediction of important quality parameters of raw lutefisk, such as water holding capacity, texture/consistency, and odour/taste characteristics. Valorization and process optimization of lutefisk as well as analytical methods used for monitoring quality of this product should be further studied in future work. 

## Figures and Tables

**Figure 1 molecules-25-01191-f001:**
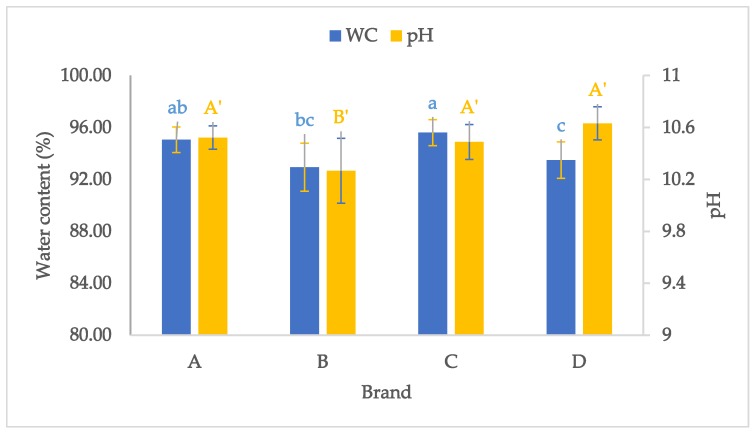
Mean values for water content (WC) and pH of the four (A, B, C, and D) lutefisk brands. The small letters and capital letters with an apostrophe indicate significant differences for the WC and pH, respectively.

**Figure 2 molecules-25-01191-f002:**
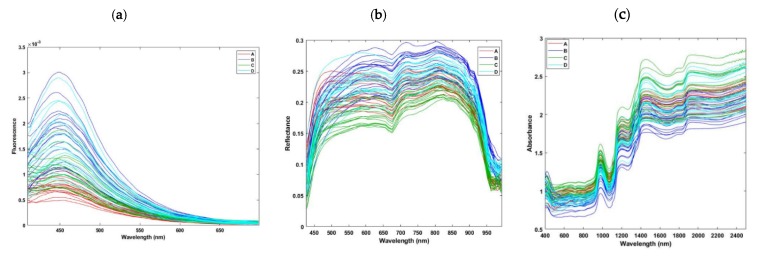
Raw spectra from online fluorescence (**a**) and diffuse reflectance (**b**) hyperspectral imaging, and offline visible/near diffuse reflectance infrared (**c**) measurements of the four lutefisk brands.

**Figure 3 molecules-25-01191-f003:**
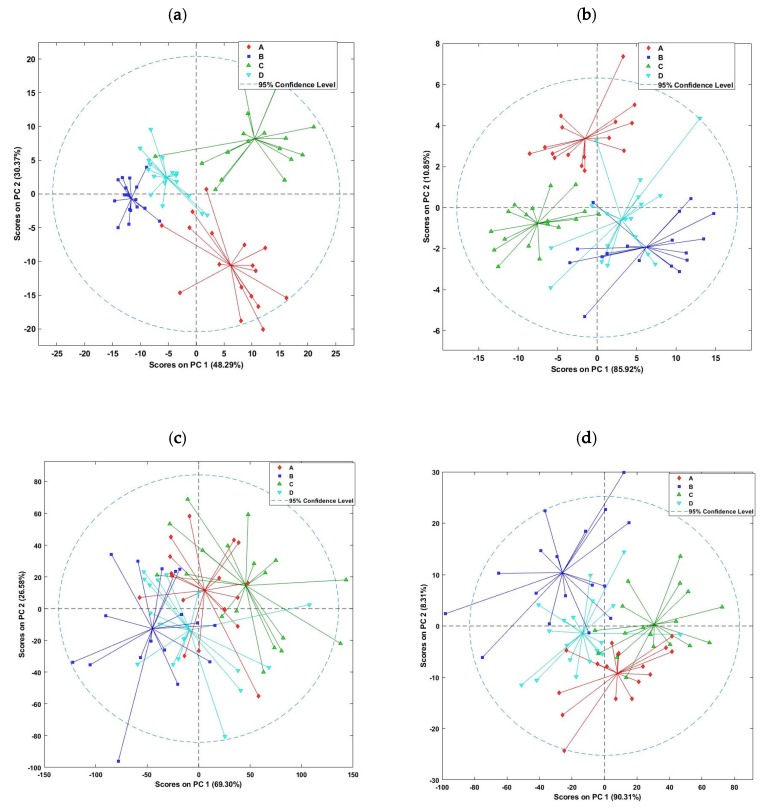
Principal component analysis (PCA) applied to the spectra obtained from online fluorescence (**a**) and diffuse reflectance (**b**) hyperspectral imaging, and offline full range (400–2500 nm) visible/near (**c**) and 400–1000 nm visible/near (**d**) infrared diffuse reflectance of the four lutefisk brands.

**Figure 4 molecules-25-01191-f004:**
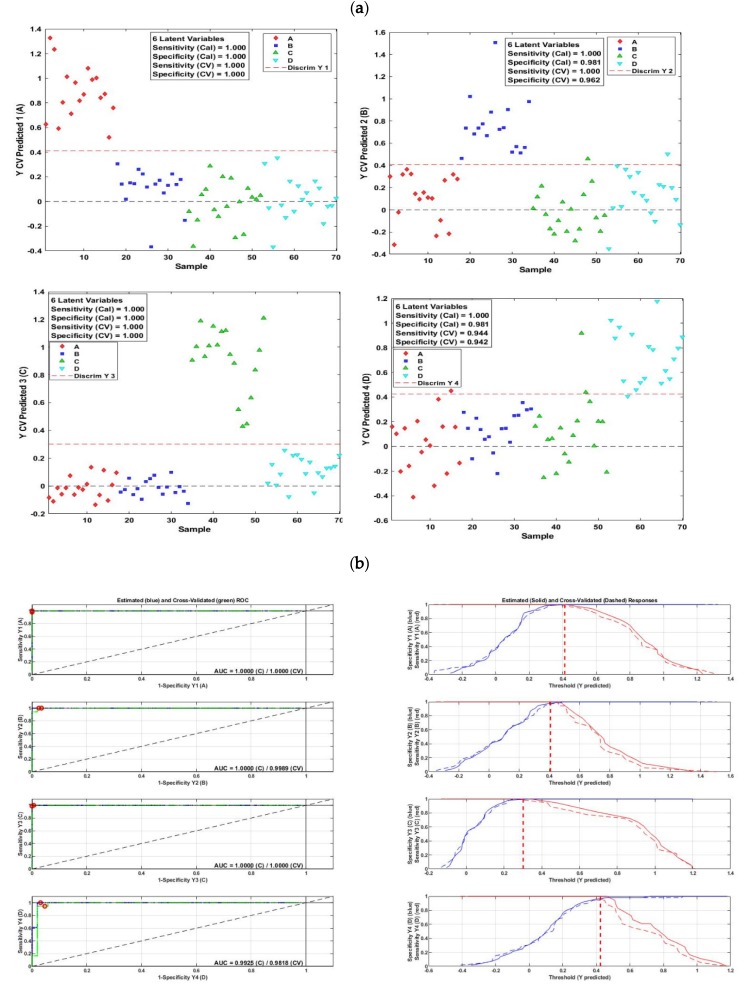
Partial least square discriminant analysis (PLS-DA) results obtained from the fluorescence measurements of the four lutefisk brands: Cross-validated predicted brand (**a**) and ROC plots (**b**).

**Figure 5 molecules-25-01191-f005:**
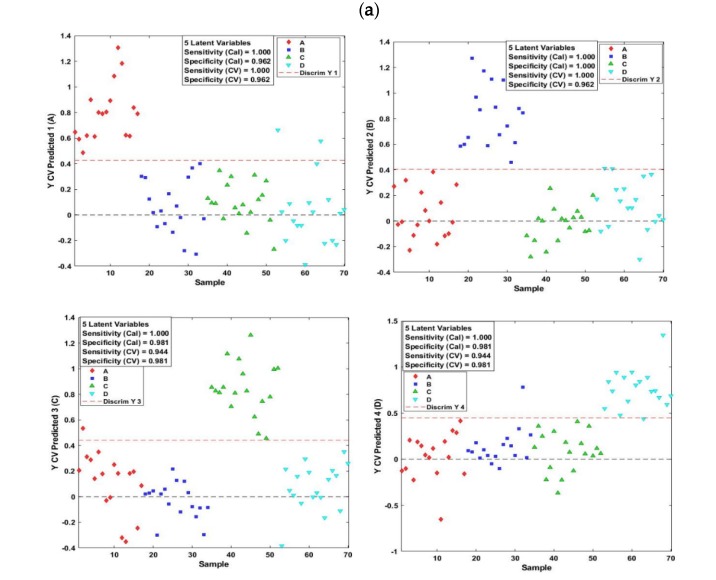
Partial least square discriminant analysis (PLS-DA) results obtained from the online diffuse reflectance hyperspectral imaging measurements of the four brands: Cross-validated predicted brand (**a**) and ROC plots (**b**).

**Figure 6 molecules-25-01191-f006:**
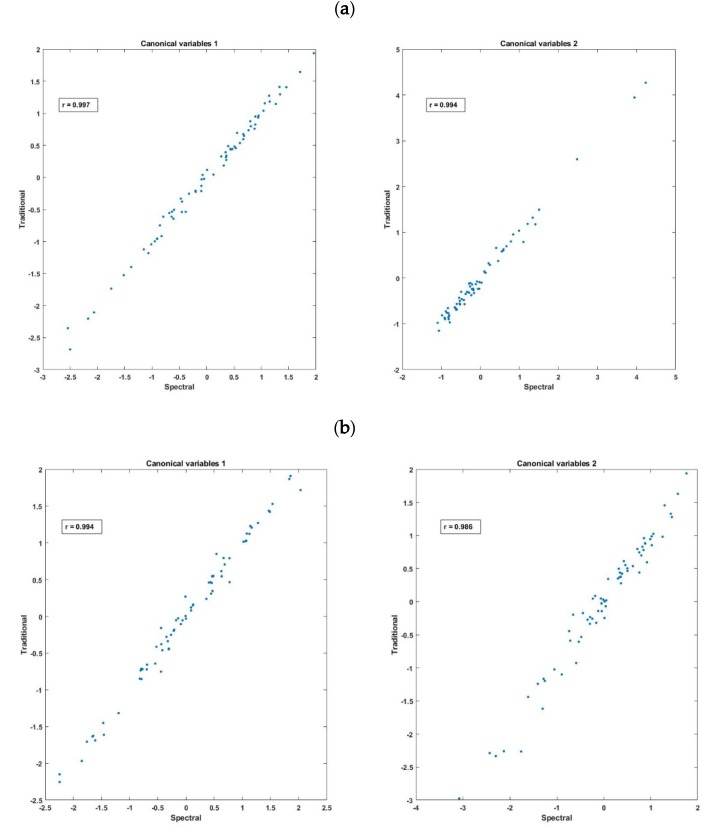
First two canonical variables calculated from the traditional measurements and (**a**) the fluorescence spectra and (**b**) the diffuse reflectance data.

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
