# Peer review of "Performance of Fluorescence and Diffuse Reflectance Hyperspectral Imaging for Characterization of Lutefisk: A Traditional Norwegian Fish Dish"

_molecules, 2020, doi:10.3390/molecules25051191_

Round 1
Reviewer 1 Report
Recommendation: Do not publish.
Comments:
This manuscript describes the feasibility of two online spectroscopic methods for the characterization of four commercially available brands of lutefisk. However, the novelty of the study is not well documented. In particular, the experimental design needs more rationale approaches to make final conclusions. Overall, the manuscript does not meet the standards of the journal and results should not be published.
Author Response
We thank the reviewer for her/his time and effort in reviewing our manuscript.
The reviewer argues that our work lacks novelty. The originality of the paper is explained in the introduction section (lines 49-51, 76-79). In fact, we think that the novelty of this paper is two-fold: First, from the material point of view, namely lutefisk. To date, only one paper has been published on this traditional food (please see reference 5). Second, from analytical point of view, in our opinion, combination of fluorescence spectroscopy with diffuse reflectance allows more detailed information on quality assessment of lutefisk.
Reviewer 2 Report
attached.

Author Response
Please find the reviewers’ comments in italic font and our responses in regular font.
The authors applied fluorescence and diffuse reflectance hyperspectral imaging to characterize lutefisk. The literature is comprehensive, the study is well conducted, and the results are clearly presented and discussed. However, before recommending applying the new techniques in industrial-scale, further validation should be carried out on a higher range of samples and by different labs in e.g. a ring study. The study can be published with very minor changes.
We would like to thank the reviewer for the constructive comments and suggestions.
- Page 1, lines 26-29: Please weaken this statement as a more comprehensive validation of the results is needed such as testing more samples, and testing reproducibility and repeatability of the analytical methods.
We agree with the suggestion made by the reviewer, and we have modified our conclusion accordingly (please see lines 26-27 and lines 463-465).
- Page 3, line 111: Please describe the meaning of the letters in the figure captions, and describe which statistical test was carried out. The figure caption should be self-explanatory.
The figure captions have been modified according to the reviewer's suggestion.
- Page 3, line 115: Spell out “TPA”, then abbreviate.
This point has been addressed in the updated version.
- Page 12, line 338: Please provide information about the temperature and humidity in the “cold room”.
This remark has been addressed in the updated version.
- Page 12, lines 349-350: Can you provide a reference for the statement made here that measurements of ground samples and intact muscles is best for pH measurements?
The two measurements provided similar results. However, we think that the direct measure of pH on intact muscle is better, as no preparation of the sample is required. It can be noticed from the scientific literature that some authors measure pH directly on fish muscles, while others prefer measurements on the homogenized samples.
We think that the other comments were made for another paper and were sent to us by error; for example, Page 1, line 18: Please remove “especially for craft brewers. There is nothing about “craft brewers” in our paper!
Reviewer 3 Report
The characterization of food using non-destructive online testing is interesting to automate the food gradation. The paper has used water contents and pH as the parameters for food gradation. The authors have done a very good job and it is evident from ROC curves of figure 4 and 5.
The work is in in its early stages and the authors could look into not only grading but also online rejecting the bacterial and viral infected food.
Author Response
Please find the reviewers’ comments in italic font and our responses in regular font.
The characterization of food using non-destructive online testing is interesting to automate the food gradation. The paper has used water contents and pH as the parameters for food gradation. The authors have done a very good job and it is evident from ROC curves of figure 4 and 5.
The work is in in its early stages and the authors could look into not only grading but also online rejecting the bacterial and viral infected food.
Thank you for the kind comments of the reviewer. We are glad to hear that our manuscript is good. We agree that microbial growth is important, and this will determine the shelf life of lutefisk. In fact, this is something that we are currently planning to look into in detail in other additional future experiments.
Reviewer 4 Report
The total number of samples and replicates must be added and explained.
The standard error obtained in the prediction must be added, plus the bias, slope.
The loadings must be added and discussed.
Authors must clarify why they made the distinction between diffuse reflectance and and VIS/NIR. The Foss system used is also using diffuse reflectance.
The validation methodology and validation of the models must be added and discussed.
The criteria used to classify the samples must be also added.
Author Response
Please find the reviewers’ comments in italic font and our responses in regular font.
Thank you for the reviewer for valuable comments and questions.
The total number of samples and replicates must be added and explained.
Information about the material used in this study, number of samples... has been added to the “Materials and Methods” section. Please see the lines 372-374.
The standard error obtained in the prediction must be added, plus the bias, slope.
Most criteria used to evaluate the performance of classification models are shown on the figures 4 and 5. The errors have been also added to the updated version (please see lines 250-252, 285-286). Concerning the canonical correlation analysis, the canonical correlation coefficients and more information about the statistics used are added to the updated version (lines 328-330).
The loadings must be added and discussed.
The loadings have been discussed for PCA performed on fluorescence data (lines 217-221) in the updated version. Loadings of the canonical correlation analysis are also discussed (lines 331-336, 342-345). We have already 6 figures in the manuscript, but if it is really necessary to add an additional figure for the loadings, so this can be added as a supplementary material.
Authors must clarify why they made the distinction between diffuse reflectance and and VIS/NIR. The Foss system used is also using diffuse reflectance.
Yes, indeed, the two measurements are based on the same principal, even though the results are expressed in different ways (in reflectance for the offline analysis and in absorbance for the online measurements). However, in this manuscript the diffuse reflectance was used mainly to describe the measurements performed online (on the conveyor belt) using hyperspectral imaging, while the visible/near infrared measurements were meant for the offline analysis (lab instrument). We agree that this can create possible confusion. So, in order to avoid any misunderstanding, we revised the manuscript on view of this suggestion, and we made the necessary corrections (for examples, lines 155-156, 171-172, 186…).
The validation methodology and validation of the models must be added and discussed.
Cross validation was employed to evaluate the performance of the classification models. More details can be found in the references 17-18. All the results presented in the figures 4 and 5 are cross-validated.
The criteria used to classify the samples must be also added.
A general description of the classification models and criteria used to evaluate the models are provided in the “Chemometric Analysis” section; lines 422-439. Some of the evaluation criteria, like sensitivity and specificity are shown on the figures 4 and 5. In addition, the ROC curves, obtained by plotting the false positive (1−specificity) vs. sensitivity values are also displayed.
Round 2
Reviewer 1 Report
Recommendation: Do not publish.
Comments:
I have read the authors' response to the critiques of the original submission and find that the manuscript is little improved or changed. The manuscript contains observations but is not a full study and the conclusions can not be justified on the basis of the rest of the paper. Again, the manuscript does not meet the standards of the journal “Molecules” and results should not be published.
Reviewer 4 Report
Accept